

# AP003352.1/miR-141-3p axis enhances the proliferation of osteosarcoma by LPAR3

Hongde Yu*, Bolun Zhang*, Lin Qi, Jian Han, Mingyang Guan, Jiaze Li and Qingtao Meng

Department of Orthopedics, Dalian NO.3 People's Hospital, Dalian, China
* These authors contributed equally to this work.

## ABSTRACT

Osteosarcoma (OS) is a highly malignant tumor with a poor prognosis and a growing incidence. LncRNAs and microRNAs control the occurrence and development process of osteosarcoma through ceRNA patterns. The LPAR3 gene is important in cancer cell proliferation, apoptosis and disease development. However, the regulatory mechanism of the ceRNA network through which LPAR3 participates in osteosarcoma has not been clarified. Herein, our study demonstrated that the AP003352.1/miR-141-3p axis drives LPAR3 expression to induce the malignant progression of osteosarcoma. First, the expression of LPAR3 is regulated by the changes in AP003352.1 and miR-141-3p. Similar to the ceRNA of miR-141-3p, AP003352.1 regulates the expression of LPAR3 through this mechanism. In addition, the regulation of AP003352.1 in malignant osteosarcoma progression depends to a certain degree on miR-141-3p. Importantly, the AP003352.1/miR-141-3p/LPAR3 axis can better serve as a multi-gene diagnostic marker for osteosarcoma. In conclusion, our research reveals a new ceRNA regulatory network, which provides a novel potential target for the diagnosis and treatment of osteosarcoma.

## INTRODUCTION

Osteosarcoma is a primary malignant bone tumor with a high mortality rate, which often occurs during adolescence and the growth period of children (*Yu et al., 2022*; *Gill & Gorlick, 2021*; *Meltzer & Helman, 2021*). In recent years, there has been a rapid rise in the trend of tumor diseases. Moreover, the cure rate of osteosarcoma patients has remained seriously low, with these patients being prone to recurrence (*Hingorani et al., 2022*; *Yang et al., 2022*). Currently, the treatment modalities for osteosarcoma primarily include surgical resection and adjuvant chemotherapy with high-dose methotrexate, cisplatin, and other drugs (*Rathore et al., 2021*). The effective therapeutic target of osteosarcoma has not been fully clarified. In addition, due to the absence of definite diagnostic markers, the early-stage diagnosis rate of patients with osteosarcoma remains poor. Most osteosarcoma patients also have drug resistance and tumor metastasis after treatment (*Shoaib, Fan & Irudayaraj, 2022*; *Zheng et al., 2022*). Therefore, it is urgent to identify new, efficient diagnostic methods and treatments to improve the prognosis of osteosarcoma and enhance the clinical diagnosis rates and cure rates of osteosarcoma patients.

Corresponding author
Qingtao Meng, spine_mqt@163.com

Non-coding RNAs refer to RNA molecules that cannot be translated into protein after transcription (*Yi et al., 2015*). Non-coding RNAs regulate cell growth and development by participating in the regulation of multiple signaling pathways, which is important in various cellular activities (*Li et al., 2021*; *Ali et al., 2021*). With the development of biomedicine, non-coding RNAs have gradually attracted increasing attention (*Wolfien et al., 2019*; *Huttenhofer & Vogel, 2006*). MicroRNAs are non-coding RNAs that have been greatly conserved in evolution, consisting of approximately 22 nucleotides (*Aure et al., 2021*; *Lee et al., 2020*; *Seal et al., 2020*). By targeting mRNA degradation and inhibiting its translation, microRNAs have the ability to affect protein expression (*Bartel, 2009*). As a key mediator in tumorigenesis, microRNAs regulate tumor proliferation, differentiation, apoptosis, and other physiological processes (*Mccall et al., 2017*; *Rupaimoole & Slack, 2017*). Abnormal expression of microRNAs is closely related to the pathogenesis of tumors and many diseases, which demonstrated that microRNAs are pivotal to homo sapiens (*So et al., 2021*; *Mitra et al., 2020*). Recently, microRNAs have been widely studied as regulators of gene expression and drug therapeutic targets of diseases. As important regulators of cancer biology, miRNAs play important roles in different sarcomas (*Kelly et al., 2013*). It seems that miRNAs can function as oncogenes or tumor suppressors *via* upregulating or downregulating miRNA expression, especially in osteosarcoma (*Gougelet et al., 2011*; *Song et al., 2009*). Compared with normal osteoblasts, there are differentially expressed miRNAs in osteosarcoma tissue, which are associated with the proliferation, metastasis, and chemotherapy resistance of osteosarcoma (*Maire et al., 2011*). MiR-141-3p can inhibit the occurrence and development processes of many tumors. According to the study of *Wang (2020)*, miR-141-3p can reduce osteosarcoma malignancy by degrading LDHB. In addition, miR-141-3p can also inhibit the growth of osteosarcoma cells *via* the regulation of EGFR, which indicates that miR-141-3p may play an inhibitory role in the development of osteosarcoma (*Wang et al., 2018*). LncRNAs are long non-coding RNAs with a length of >200 nucleotides, which wildly regulate cellular processes (*Luo et al., 2021*; *Jiang et al., 2019*; *Kim et al., 2018*). LncRNAs have gradually become key regulators of tumor biological behavior, which regulate the development of tumors through various mechanisms (*Marney et al., 2021*; *Li et al., 2018*). LncRNAs are important in gene regulation, functioning through multiple mechanisms (*Liu et al., 2021*). For example, LncRNAs could function as ceRNA in competitive combination with miRNAs, thereby regulating the expression of miRNA (*Li et al., 2020*). LncRNAs are related to the physiological and pathological processes of tumors (*Wee et al., 2012*; *Franco-Zorrilla et al., 2007*). An increasing number of studies showed that lncRNAs regulated tumor cell apoptosis, epithelial mesenchymal transformation, tumor drug resistance, and other mechanisms (*Cazalla & Steitz, 2010*; *Wang et al., 2020*). Targeting lncRNAs has become a promising step by step treatment for tumors. Previous studies showed that some distinctly expressed lncRNAs in osteosarcoma are related to the development of osteosarcoma (*Xun et al., 2021*). It is reported that LncRNA PCAT6 act as an miRNA sponge to control osteosarcoma proliferation, migration and invasion (*Zhu et al., 2020*; *Guan, Xu & Zheng, 2020*). *Pan et al. (2018)* showed that lncRNA FBXL19-AS1 spongy miR-346 regulated the proliferation of osteosarcoma cells. LncRNAs have become markers of poor prognosis in

osteosarcoma patients. Moreover, research has shown that AP003352.1 is closely related to the malignant progression and could result in significant clinicopathological characteristics. AP003352.1 is superior to other lncRNAs in predicting overall survival, which could predict the overall prognosis and establish the prognostic characteristics of cancer patients (*Xuan et al., 2021*). However, the specific mechanism of AP003352.1 in osteosarcoma has not been clarified.

MicroRNAs can induce gene silence by binding to mRNA, while competitive endogenous RNAs (ceRNA) can affect gene expression by reacting to microRNA in a competitive manner (*Karreth & Pandolfi, 2013*). CeRNA is important in the pathogenesis of cancer, serving as a biomarker for diagnosis and a potential therapeutic target for cancer (*Qi et al., 2015*). For example, ceRNA network has significance in DXR-resistant breast cancer cells (*Gao et al., 2017*). In addition, ceRNA network also regulates the proliferation of prostate cancer cells (*Gao et al., 2019*). *Zhu et al. (2019)* reported that ceRNA can participate in osteosarcoma chemoresistance. However, the ceRNA regulatory network in osteosarcoma remains to be investigated. Lysophosphatidic acid receptor 3 (LPAR3) is the receptor of lysophosphatidic acid (LPA) that participates in the physiological processes of various cells in the body (*Xia & Jie, 2020*). During embryonic development, LPAR3 was involved in the correct formation of various organs (*Lin et al., 2020*). Many studies proved that LPAR3 played a pivotal role in the development of cancer. As a regulator, LPAR3 participates in regulating malignant pathways and affects the proliferation and apoptosis of cancer cells (*Li et al., 2019*). During the progression of osteosarcoma, LPAR3 can regulate the effect of osteosarcoma cells through the LPA signal pathway (*Minami et al., 2021*; *Takahashi et al., 2018*). In osteosarcoma cells with LPAR3 knock down, the migration ability and motile activities of cells decreased significantly (*Tanabe et al., 2012*; *Kurisu et al., 2022*). LPAR3 showed many malignant characteristics in osteosarcoma, but the mechanism through which non-coding RNA regulates LPAR3 in osteosarcoma has not been clarified.

In this study, we found that, as the ceRNA of miR-141-3p, AP003352.1 regulated the expression of LPAR3 in osteosarcoma. First, we identified the regulatory mechanism of AP003352.1, miR-141-3p, and LPAR3 in the malignant development of osteosarcoma. Then, we predicted that AP003352.1 could bind to miR-141-3p and regulated LPAR3 as a molecular sponge. In addition, AP003352.1/miR-141-3p axis could simultaneously regulate the proliferation of osteosarcoma through LPAR3. Finally, we verified the feasibility of using the AP003352.1/miR-141-3p/LPAR3 axis as a biomarker.

# METHODS

## Data collection

The mRNA and lncRNA sequencing data of 88 patients with osteosarcoma and their corresponding clinical information were downloaded from the TCGA database as cluster analysis data and COX analysis data. The miRNA sequencing data of 65 patients with osteosarcoma and their corresponding clinical information were downloaded from the GSE39040 GEO database as cluster analysis data and COX analysis data. The TARGET-OS data downloaded from the TCGA database was used as the training cohort, while the

TCGA-sarc data downloaded from TCGA database was used as the validation cohort. The official websites of TCGA database and GEO database are as follows:

TCGA: https://www.cancer.gov/ccg/research/genome-sequencing/tcga.

GEO: https://www.ncbi.nlm.nih.gov/geo/.

## Univariate COX analysis

Univariate COX analysis was applied to evaluate the overall survival rate and status of osteosarcoma patients in the TCGA database, which was then applied to assess the prognostic value of the mRNAs, miRNAs, and lncRNAs. Univariate COX analysis was used to assess the prognostic value of risk scores, deriving hazard ratios (HR) and 95% confidence intervals (CI) for each variable.

## Differential expression analysis

Before differentially expressed gene (DEG) identification, the data were normalized using the limma package (3.52.4). Meanwhile, the data type used for the DEG identification was the new TPM value after log2 transformation. According to the new standard of log FC $\geq 1$ and $p < 0.05$ for screening the condition, the DEGs between osteosarcoma tumor samples and precancerous samples were analyzed, and the obtained results could be used for unicox analysis and LASSO screening.

## Gene enrichment analysis

Gene enrichment analysis is an analytical method used to analyze the expression and enrichment of DEGs. The R package clusterProfiler (4.4.4) was used to analyze the enrichment of intersecting genes on different signaling pathways. $P$-value $< 0.05$ indicated that the enrichment genes in corresponding pathways were significant.

## Construction of the lncRNA-miRNA-mRNA ceRNA network

The ceRNA network of lncRNA-miRNA-mRNA was created by predicting the correlation between lncRNA, miRNA, and mRNA. miRNA is a key component of the ceRNA network, which can connect with lncRNA and mRNA. The R package multiMiR (1.18.0) was used to analyze the ceRNA network. In this study, the network was visualized by Cytoscape software.

## Cell lines and cell culture

HEK293T, U2OS and MG63 cell lines were purchased from ATCC. They were cultured in DMEM with 10% FBS, 100 μg/mL penicillin, and 100 μg/mL streptomycin. Meanwhile, HEK293T, U2OS and MG63 cells were incubated in a humidified incubator with 5% $CO_2$ at 37 °C.

## CCK8 assay

CCK8 assay was performed to test the cell viability. A total treated of $5 \times 10^3$ U2OS cells were inoculated into 96-well culture plates. After 24 h of cell culture, CCK-8 was added into the plates with 10 μL per well 3 h before testing. Afterwards, the absorbance value at

was measured 450 nm with an enzyme marker. Each group of experiments was repeated for three times.

## Clone formation assay

To clarify the effect of ceRNA axis on clone formation, single cells of U2OS were cultured in six-well plates with $1 \times 10^3$ cells per well. Withing 10 days of incubation, the colonies were clearly visible, and polyformaldehyde was used to fix cells, while crystal violet was used for cell staining.

## Western blot

Cells were lysed using RIPA lysis buffer, and the BCA kit was used for cell quantification. The protein was separated by SDS-PAGE and then transferred to the NC membrane. After blocking with 5% skimmed milk, the NC membrane was incubated with primary antibody at 4 °C for 14 h. TBST was used to wash the NC membrane three times, followed by sealing with the secondary antibody for 2 h. Finally, the protein expression was observed under electrophoresis gel imaging. The antibodies used include: LPAR3 (Invitrogen, Waltham, MA, USA; PA5-27074; 1:2,000) and α-Tubulin (CST, Tarzana, CA, USA; 2148S; 1: 1,000).

## RT-qPCR assay

RNAiso Plus was used to extract total RNA from the U2OS cells. HiScript III RT SuperMix qPCR kit was used to synthesize the cDNA. The expression of lncRNA relative to mRNA was determined using the ratio of sample transcripts. The primers used are shown below:

LPAR3 forward: 5′-TCGCTTACGTGTTCCTGATG-3′
LPAR3 reverse: 5′-TTCCACAGCAATAACCAGCA-3′
has-miR-141-3p: 5′-TAACACTGTCTGGTAAAGATGG-3′
AP003352.1 forward: 5′-TGCCTCAGCCTCTCAAGTAG-3′
AP003352.1 reverse: 5′-CGTGGCTCACACCTGTAATC-3′

## Dual-luciferase reporter assay

PGL4.15 was performed to construct plasmids containing the AP003352.1-wild type and mutant. U2OS cells were inoculated onto the 24-well plate. After cell adhesion, wild type and mutant AP003352.1 were transfected into U2OS with lipofectamine 3000 (Invitrogen, Waltham, MA, USA; L3000001). After transfection, luciferase activity was detected according to the specification of the Dual-Luciferase Reporter Assay System Kit (Promega, Madison, WI, USA; E1910).

## Transfection

The inhibitors and mimics of has-miR-141-3p were bought from Biomics (Nantong, China). The ASOs (Antisense Oligonucleotide) of lncRNAs AL358332.1, AC124798.1, LINC02280, AP003352.1, AC073487.1 and AC084855.2 were purchased from RiboBio (Guangzhou, China). The inhibitors and mimics of miR-141-3p and the ASOs of lncRNAs were transfected into U2OS cells using Lipofectamine 3000 (Invitrogen, Waltham, MA, USA; L3000001).
## Construction of shRNAs

The shRNAs of LPAR3 were constructed on the PLKO.1 plasmid. PSPAX2, PMD2G and shLPAR3 plasmids were transfected into HEK293T cells. After 48 h, the supernatant was collected and filtered through a 0.45 um filter membrane. Next, U2OS and MG63 cells were infected with virus solution, and 48 h later, fresh medium was replaced and 5 mg/mL puromycin was added to screen stably transfected cell lines. The sequences of shRNAs are shown below:

shLPAR3-1

5′-CCGGAATACATAGGCAATTCCAGCGCTCGAGCGCTGGAATTGCCTATGTA TTTTTTTG-3′

shLPAR3-2

5′-CCGGCAGTACATAGAGGATAGTATTCTCGAGAATACTATCCTCTATGTAC TGTTTTTG-3′

## Pearson's correlation coefficient

Pearson's correlation coefficient is used to determine the correlation between two variables, and the value of the Pearson correlation coefficient varies from −1 to 1. The larger the absolute value of the Pearson correlation coefficient, the stronger the correlation between the two variables. A correlation coefficient between close to 1 or −1 indicates that the correlation between the two variables is strong, whereas a correlation coefficient close to 0 indicates that the two variables have no correlation.

## Construction and validation of a prognostic model

The expression data of patients with OS from TCGA were treated as the training set, and the expression data of patients with sarcoma from TCGA were treated as the test set. The training set was used to build a prediction model through LASSO COX regression analysis. The risk score model obtained was as follows: risk score = (LPAR3 × 0.3996556 + AP003352.1 × 1.4405687). The test set was divided into low-risk and high-risk score groups based on the median value of the risk score. Receiver operating characteristic (ROC) curves were established to validate this model. The R package glmnet (4.1.6) was used for LASSO analysis.

## Statistical analysis

Statistical analyses were performed using R v4.2.1. The predictive ability of the model was determined by ROC curve analysis. KM curves were used to compare the difference in survival between the two risk groups. GraphPad Prism software was used for the analysis of the assay results.

# RESULTS

## Screening of lncRNAs, microRNAs, and mRNAs related to osteo-sarcoma progression

During OS tumorigenesis and development, the expression level of tumor markers changes significantly. A majority of tumor markers will be used as oncogenes and antioncogenes to

regulate the malignant progression of tumors. However, the exact role of the ceRNA axis, comprising of these genes, in osteosarcoma has not yet been clarified. To explore the internal mechanism, we performed differential expression analysis and univariate Cox analysis on mRNAs, lncRNAs, and miRNAs of osteosarcoma samples and precancerous samples, respectively. The results showed that there were 4,462 differentially expressed genes in mRNAs (up regulated genes: 2,333; down regulated genes: 2,129) and 1,242 survival related genes (oncogenes: 518; tumor suppressor genes: 724). A total of 2,348 differentially expressed lncRNAs (up regulated genes: 1,208; down regulated genes: 1,140) and 562 survival related genes (oncogenes: 319; tumor suppressor genes: 243) were detected. A total of 68 differentially expressed genes were found in microRNAs (upregulated genes: 37; downregulated genes: 31). The PCA diagrams showed the dispersion of differential analysis (Fig. S1A). The upregulated oncogenes and downregulated suppressor genes in osteosarcoma were extracted after obtaining the overlapping differentially expressed genes and survival related genes, including 186 protein-coding mRNAs and 45 lncRNAs (Figs. 1A–1C). The Venn diagrams showed the overlapped genes (Fig. S1B).

## Enrichment of tumor progression related genes in osteosarcoma

To further explore the importance of genes in osteosarcoma, we performed GSEA enrichment analysis of mRNAs. The C5-GO enrichment analysis results demonstrated that most of the genes were involved in the malignant progression of osteosarcoma, including the proliferation function (Figs. 2A and 2B). Osteosarcoma is a highly malignant tumor originating from the bone marrow, which is characterized by the proliferation of osteoblastic precursor cells and the production of osteoid or immature bone. Traditional chemotherapy is the first choice for drug treatment of osteosarcoma. Chemotherapy drugs can significantly inhibit the proliferation of osteosarcoma cells, which also indicates that the inhibition of proliferation is the primary treatment for osteosarcoma. At the same time, the proliferation related pathways mTOR and MAPK signaling pathways play an pivotal role in the occurrence and development of osteosarcoma. In conclusion, the proliferation phenotype plays an important role in the development of osteosarcoma, so our research focuses on the proliferation pathway in the GO term analysis.

## The construction of ceRNA network in osteosarcoma

According to the above results, the mRNAs, lncRNAs, and microRNAs obtained from survival analysis were used to construct ceRNA networks. The networks regulated the malignant progression of osteosarcoma, including 41 lncRNAs, 8 miRNAs, 36 mRNAs, and 85 groups of interaction axes (Fig. 3A). Cytoscape software was used to draw a network diagram of the genes regulation (Fig. S2). Among mRNAs, LPAR3 is important in the regulation of osteosarcoma. As a regulatory factor, LPAR3 participated in the regulation of the cell signal pathway, which could affect the proliferation and apoptosis of osteosarcoma cells. Therefore, the focus of our research was related to the ceRNA axis of LPAR3. The results showed that the AL358332.1/miR-141-3p, AC124798.1/miR-141-3p, LINC02280/miR-141-3p, AL135960.1/miR-141-3p, AP003352.1/miR-141-3p,

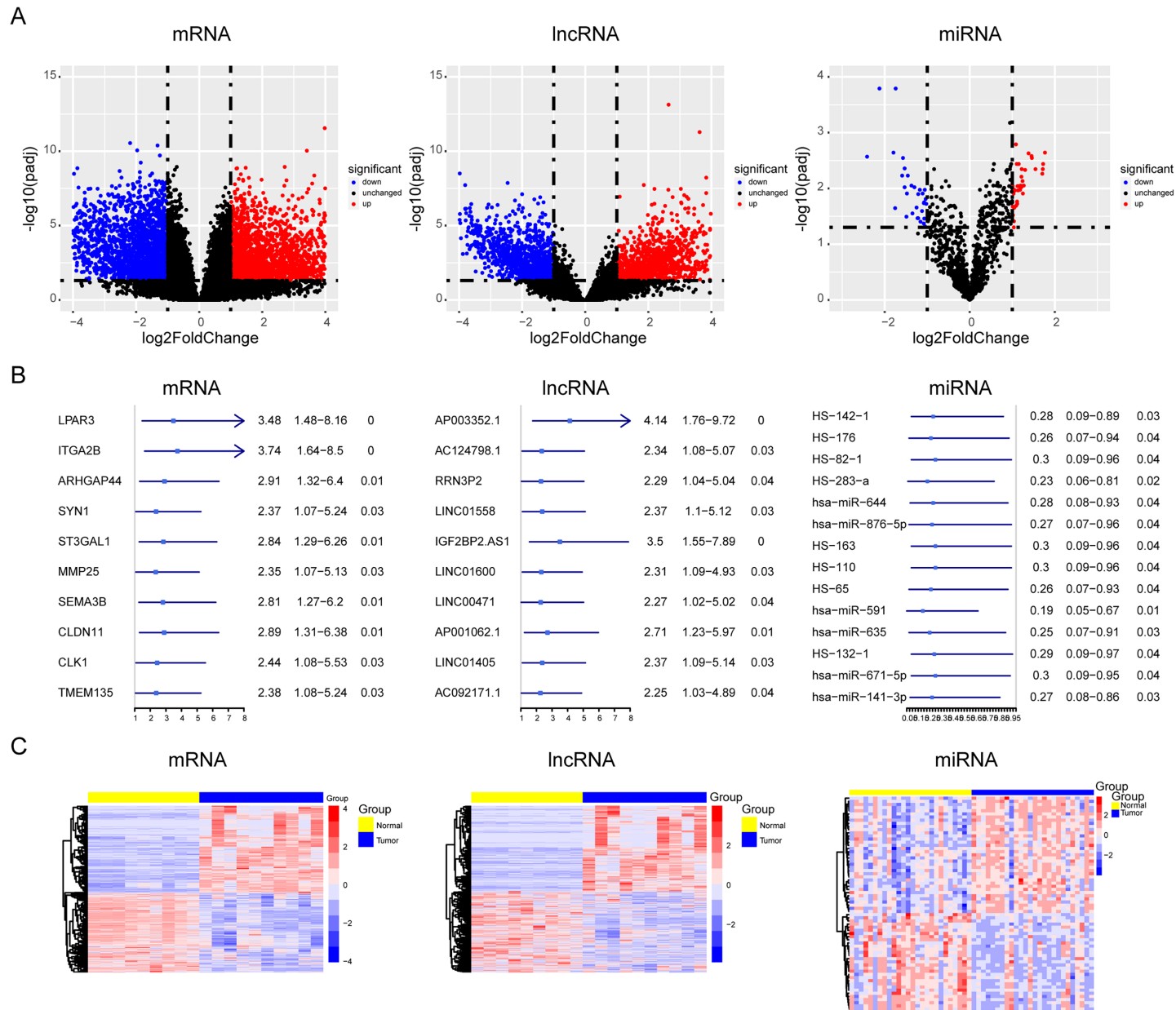

**Figure 1 Screening of lncRNAs, microRNAs, and mRNAs related to osteosarcoma progression.** The results of univariate cox and differential expression analysis of mRNAs, lncRNAs and miRNAs from the TCGA database (Target_OS) and GSE39040 database. Volcano plots of differential expression analysis of mRNAs, lncRNAs and miRNAs (A). The top ten results of univariate cox in mRNAs, lncRNAs and miRNAs (B). Heatmaps of overlapped results between univariate cox and differential expression analysis in mRNAs, lncRNAs and miRNAs (C).

AC073487.1/miR-141-3p, and AC084855.2/miR-141-3p axes could regulate LPAR3 (Fig. 3B).

## AP003352.1 may regulate LPAR3 at the mRNA and protein levels

To explore the main lncRNA that regulated LPAR3, differential expression analysis and survival analysis were conducted on AL358332.1, AC124798.1, LINC02280, AP003352.1,
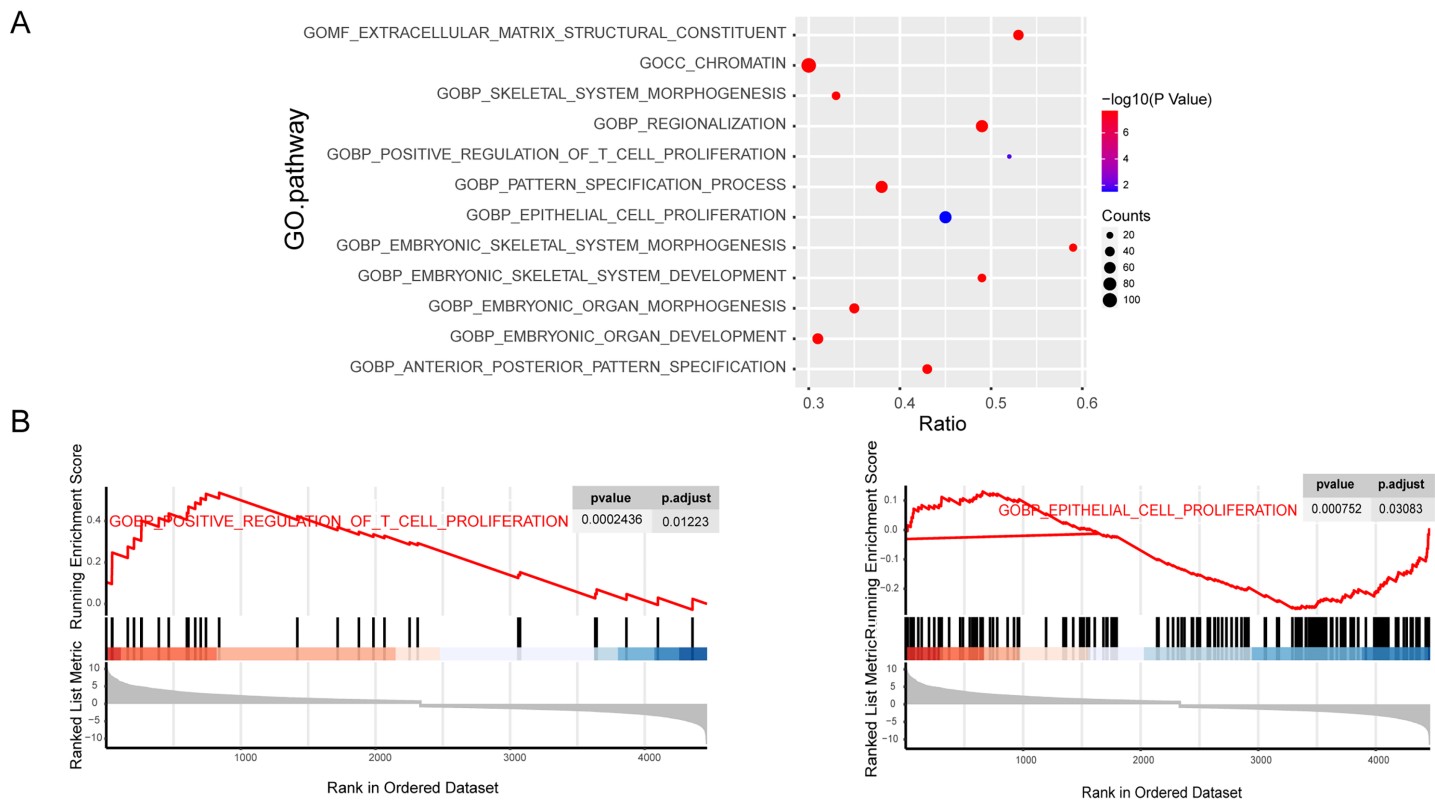

**Figure 2 Enrichment of tumor progression related genes in osteosarcoma.** C5 GO fuctional enrichment analysis of the diferentally exressed mRNAs. Bubble diagram of C5 GO analysis of mRNAs (A). C5 GO enrichment analysis of regulated mRNAs (B).

AC073487.1, and AC084855.2, respectively (The count of AL135960.1 is too low). The results of differential expression analysis showed that four types of lncRNAs were significantly overexpressed in osteosarcoma (Fig. 4A). Therefore, these lncRNAs may be responsible for the overexpression of LPAR3 in osteosarcoma. In addition, the results of the univariate Cox regression analysis showed that the six lncRNAs were associated with a worse survival (Fig. 4B). Next, we conducted Pearson correlation analysis on six genes. And the results indicated that only AP003352.1 and AC084855.2 had significant correlations with LPAR3 (Fig. 4C). The AUC curve showed that the survival sensitivity of lncRNA AP003352.1 was significant (Fig. S3). To further screen the results, ASO was used to knock down AL358332.1, AC124798.1, LINC02280, AP003352.1, AC073487.1, and AC084855.2, respectively. Subsequently, the changes in LPAR3 caused by knocking down the relevant lncRNA were detected at the mRNA and protein levels (Figs. 4D, and 4E). The results revealed that the expression of LPAR3 was particularly regulated by AP003352.1, while no statistical difference was observed when other lncRNAs were knocked down. Thus, AP003352.1 may be the main lncRNA regulating LPAR3.

A

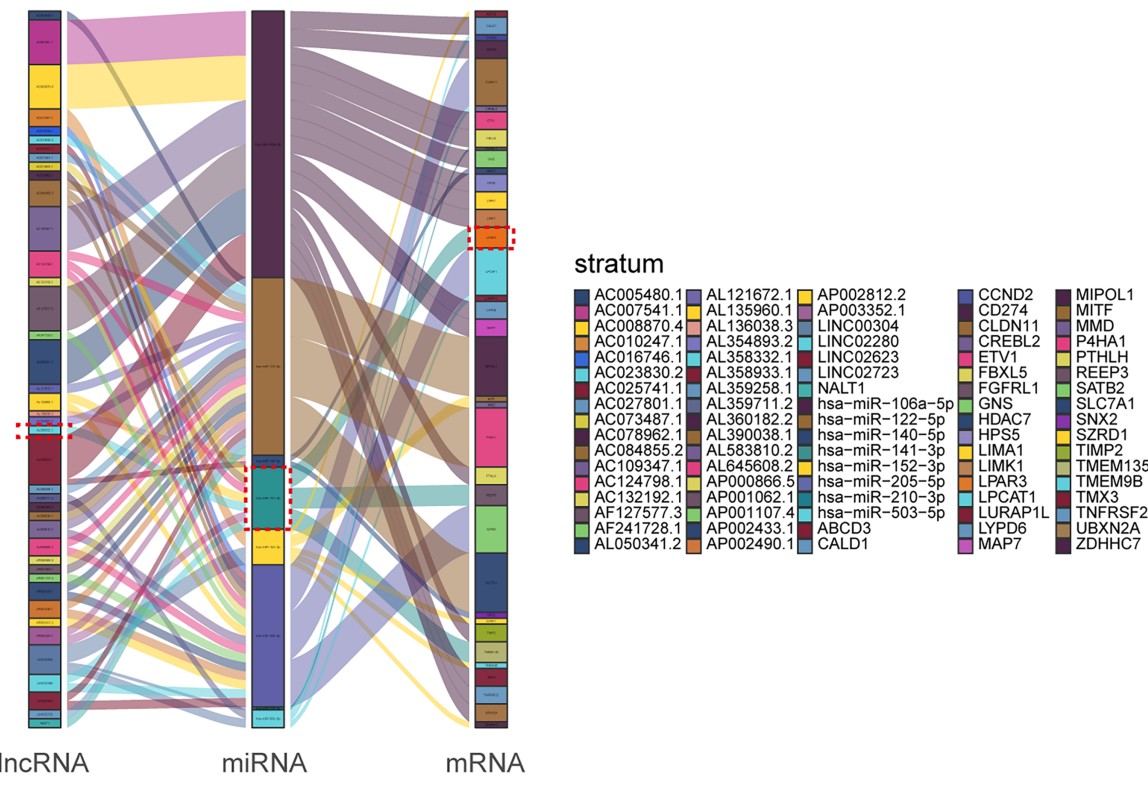

B

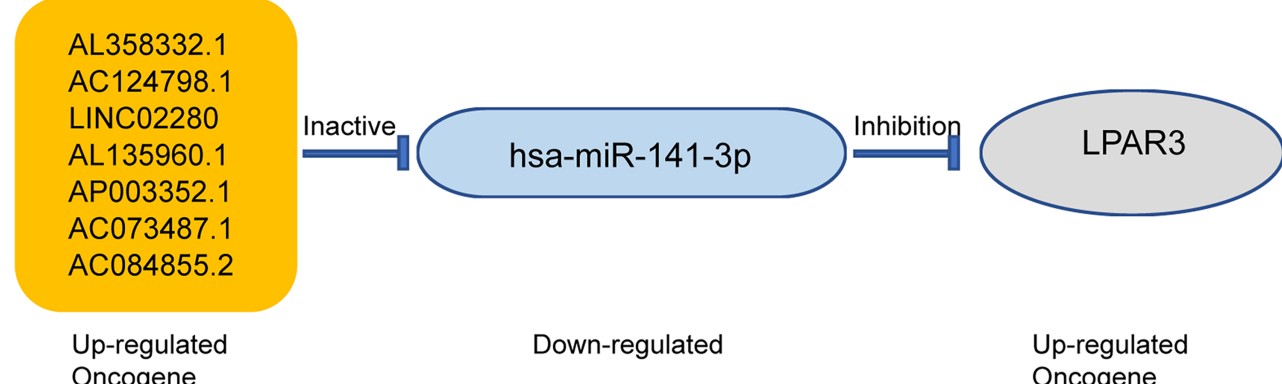

**Figure 3 The construction of ceRNA network in osteosarcoma.** Construction of osteosarcoma related ceRNA network by integrated analysis. LncRNA-miRNA-mRNA regulatory axes extracted from this ceRNA network (A). The LncRNA-miRNA-mRNA pattern diagram (B).

## AP003352.1, miR-141-3p, and LPAR3 regulate the malignant progression of osteosarcoma

According to the prediction results reported above, AP003352.1/miR-141-3p/LPAR3 may be the ceRNA axis that regulated the osteosarcoma development. However, the exact role of the ceRNA axis in the malignant progression of osteosarcoma has not yet been clarified.

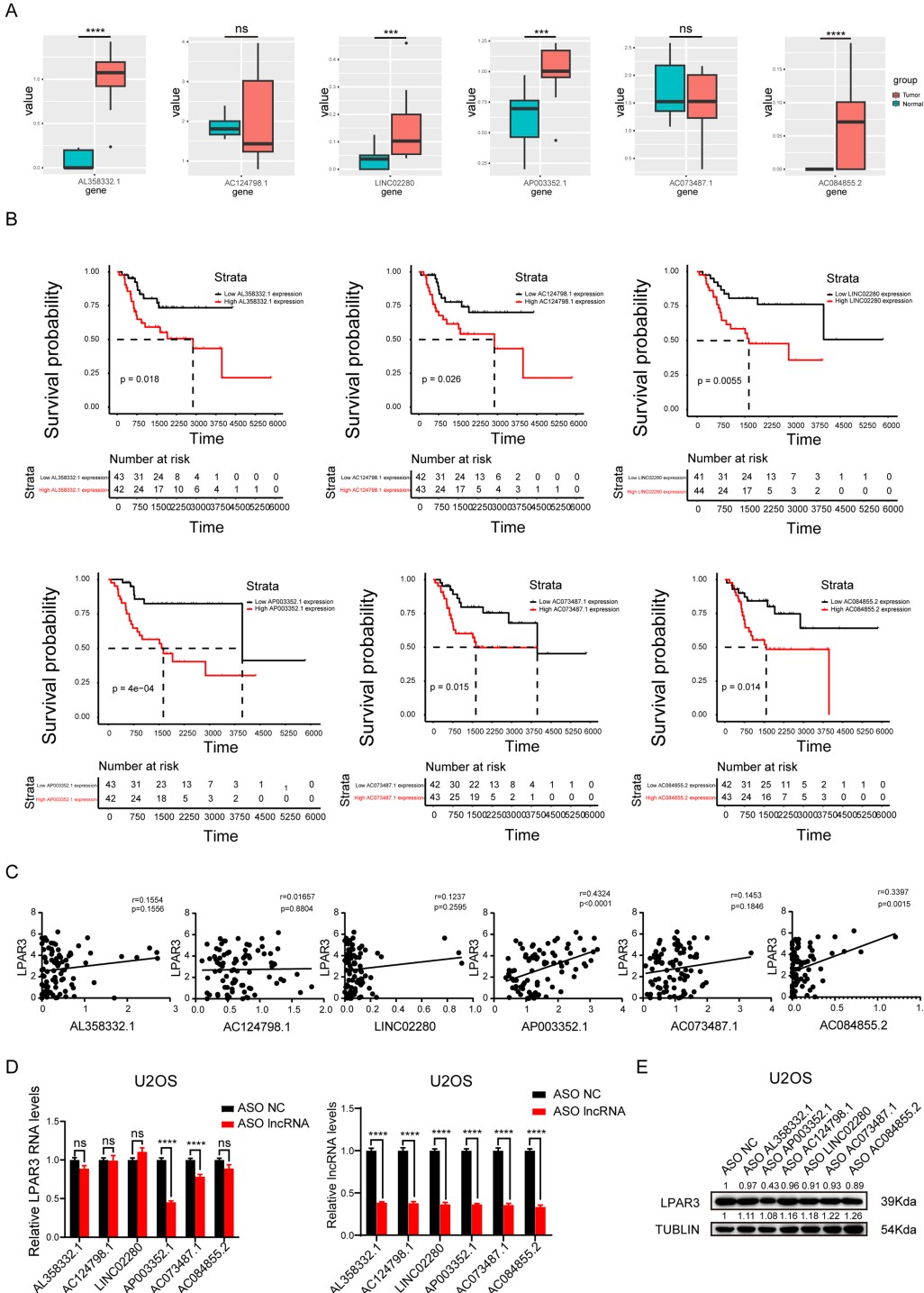

**Figure 4 AP003352.1 may regulate LPAR3 at the mRNA and protein levels.** The expression of potential lncRNAs between tumor and normal tissues in osteosarcoma patients. The red box indicates the tumor tissue and the green box indicates the normal tissue. The statistical method is T-test (A). The survival curves of potential lncRNAs in osteosarcoma patients. Black lines indicate low expression of lncRNA and red lines indicate high expression of lncRNA (B). Pearson correlation coefficient analysis between AP003352.1 and LPAR3 (C). The mRNA level of LPAR3 in U2OS cells with six potential lncRNAs knockdown by ASO. The statistical method is T-test (D). The protein level of LPAR3 in U2OS

**Figure 4** (continued)
cells with six potential lncRNAs knockdown by ASO (E). Values were expressed as the means ± SD from three experiments, and the asterisk indicates the statistical significance compared to the controls (***$p < 0.001$, ****$p < 0.0001$).   

Therefore, osteosarcoma samples were divided into two groups for differential expression analysis: LPAR3 high expression group and LPAR3 low expression group (AP003352.1 was for the same analysis). The results of differential expression analysis showed that there were many genes related to osteosarcoma proliferation (Figs. 5A and S4). To further evaluate the effect of the three genes on the proliferation of osteosarcoma, CCK8 assay and clone formation were conducted after respectively inhibiting the three genes. When the expression of AP003352.1, miR-141-3p and LPAR3 genes were changed, the value of OD450 detected by CCK8 and the number of clones were regulated accordingly. Therefore, the three genes can regulate the proliferation of osteosarcoma cells U2OS and MG63. The experimental results were consistent with the differential expression analysis results (Figs. 5B and 5C). At the end of the cloning experiment, we detected the mRNA levels of AP003352.1, miR-131-3p and LPAR3 by qPCR and the knockdown effects still existed (Fig. 5D). The above results indicated that all three genes could regulate the malignant progression of osteosarcoma.

## AP003352.1 and miR-141-3p can regulate LPAR3 in osteosarcoma

To further explore the relationship between AP003352.1, miR-141-3p, and LPAR3, ASO was used to knock down AP003352.1, and the changes in miR-141-3p and LPAR3 were detected. The results showed that LPAR3 was typically downregulated, while miR-141-3p was upregulated (Fig. 6A). Then, after altering the expression of miR-141-3p by using overexpression mimics and inhibition inhibitors, we detected the changes in the expression of AP003352.1 and LPAR3. The results indicated that AP003352.1 and LPAR3 were negatively regulated by miR-141-3p (Figs. 6B and 6C). All the above results indicated that AP003352.1 and miR-141-3p can regulate the expression of LPAR3 in osteosarcoma, which was consistent with the trend of the ceRNA axis.

## As the ceRNA of miR-141-3p, AP003352.1 regulates LPAR3 to affect the malignant progression of osteosarcoma

To determine whether AP003352.1 plays the same role as the ceRNA of miR-141-3p, the predicted binding sites of AP003352.1 and miR-141-3p were mutated complementarily (Fig. 7A). Luciferase reporter gene assay was utilized to identify the influence of miR-141-3p mutation on luciferase activity. The results showed that the luciferase activity of wild-type AP003352.1 was regulated by miR-141-3p, whereas that of the mutant-type AP003352.1 was not statistically significant (Fig. 7B). Then, we jointly predicted the enrichment of genes regulated by AP003352.1 and LPAR3 in the malignant progression pathway of osteosarcoma. The results showed that there was significant gene enrichment in the proliferation pathway (Fig. S5). Although, the genes were related to proliferation pathways in other cells, which suggested that the genes were indirectly related to

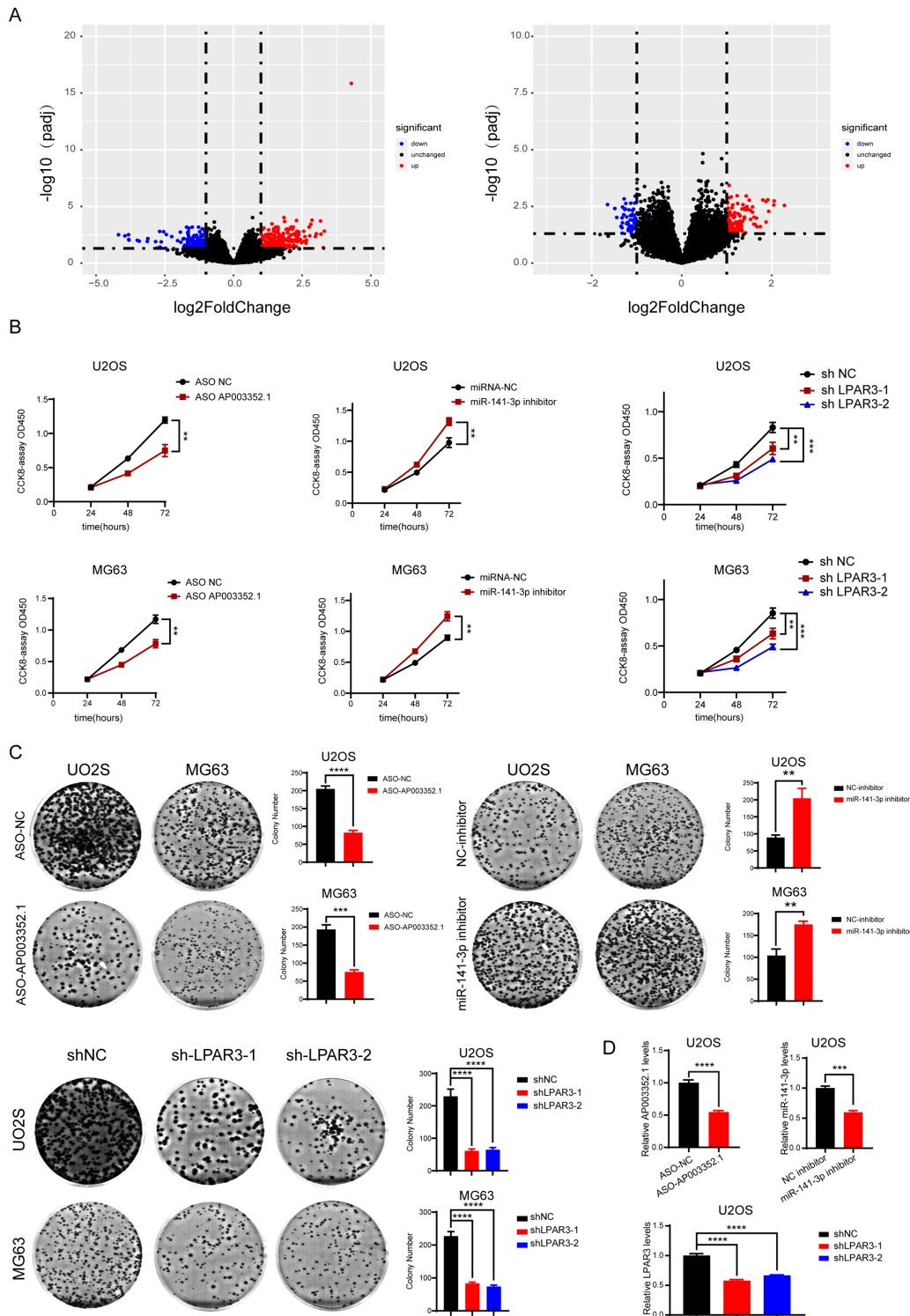

**Figure 5  AP003352.1, miR-141-3p, and LPAR3 regulate the malignant progression of osteosarcoma.**
The volcano plots of the DEGs in low expression group and high expression group among AP003352.1
and LPAR3 (A). The CCK8 assays of U2OS cells and MG63 cells were performed when AP003352.1,
miR-141-3p and LPAR3 knockdown or inhibited for 48 h (B). The cloneformationassays of U2OS cells
and MG63 cells were performed when AP003352.1, miR-141-3p and LPAR3 knockdown or inhibited for
48 h (C). The mRNA levels of AP003352.1, miR-141-3p and LPAR3 knockdown or inhibited after 10

Peer J

Figure 5 (continued)
days of clone formation (D). Values were expressed as the means ± SD from three experiments, and the asterisk indicates the statistical significance compared to the controls (**$p < 0.01$, ***$p < 0.001$, ****$p < 0.0001$).

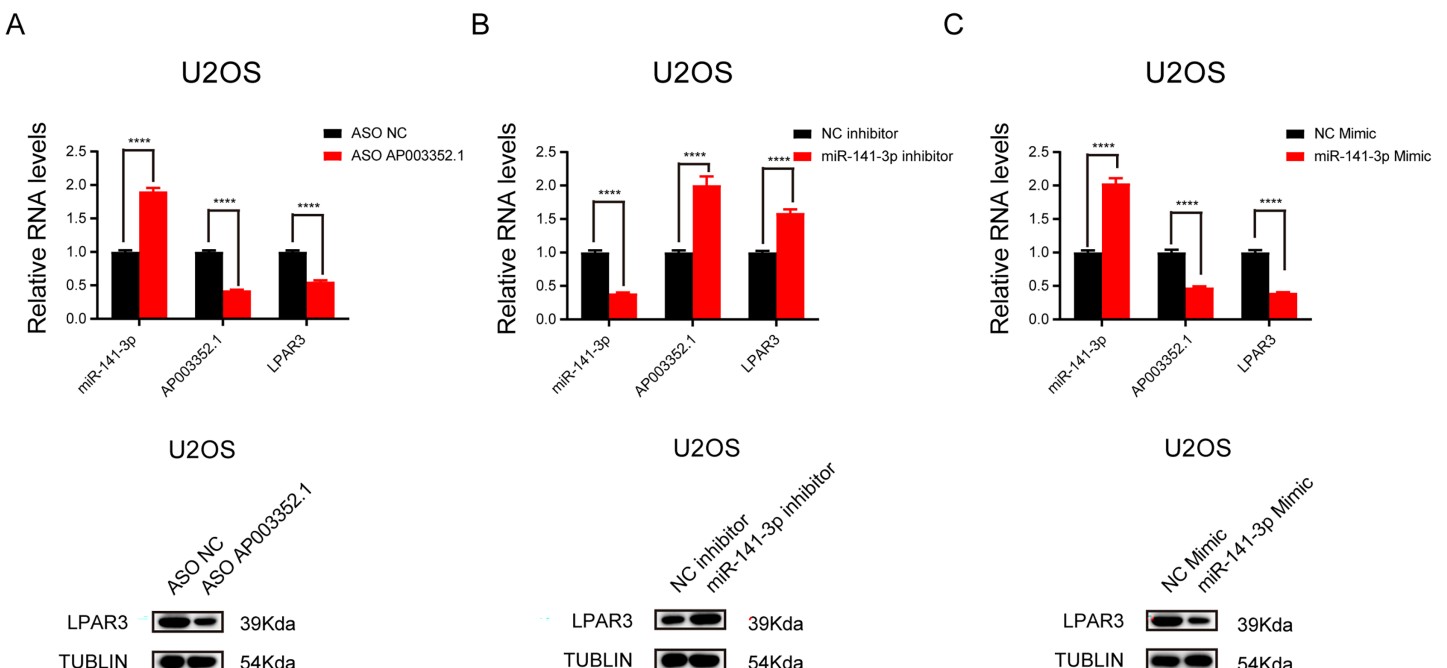

Figure 6 **AP003352.1 and miR-141-3p can regulate LPAR3 in osteosarcoma.** RT-qPCR assays and western blot were used to detect the expression of AP003352.1, miR-141-3p and LPAR3 in U2OS cells treated with AP003352.1 ASO for 48 h (A). RT-qPCR assays and western blot were used to detect the expression of AP003352.1, miR-141-3p and LPAR3 in U2OS cells treated with miR-141-3p inhibitor for 48 h (B). RT-qPCR assays and western blot were used to detect the expression of AP003352.1, miR-141-3p and LPAR3 in U2OS cells treated with miR-141-3p mimic for 48 h (C). Values were expressed as the means ± SD from three experiments, and the asterisk indicates the statistical significance compared to the controls (****$p < 0.0001$).

osteosarcoma proliferation and had certain significance. The osteosarcoma cells and other types of cells may have some of the same proliferation regulation mechanism. To verify the results, we evaluated the cell proliferation in the control group and the osteosarcoma cells using miR-141-3p mimic after knocking down AP003352.1. The results of CCK8 assays showed that in the control group, AP003352.1 knockdown inhibited the proliferation of osteosarcoma cells U2OS, while this inhibition was not observed in the mimic group (Fig. 7C). Besides, the results of clone formation assays indicated that AP003352.1 knockdown also inhibited the colony formation ability of osteosarcoma cells U2OS in the control group and in the mimic group, the inhibition of colony formation ability was disappeared (Fig. 7D). Besides, we also detected the cell proliferation in the control group and the osteosarcoma cells after knocking down LPAR3. The results showed that AP003352.1 knockdown could inhibit the proliferation of osteosarcoma cells U2OS in the control group, but the inhibitory effect was not observed in LPAR3 knockdown group

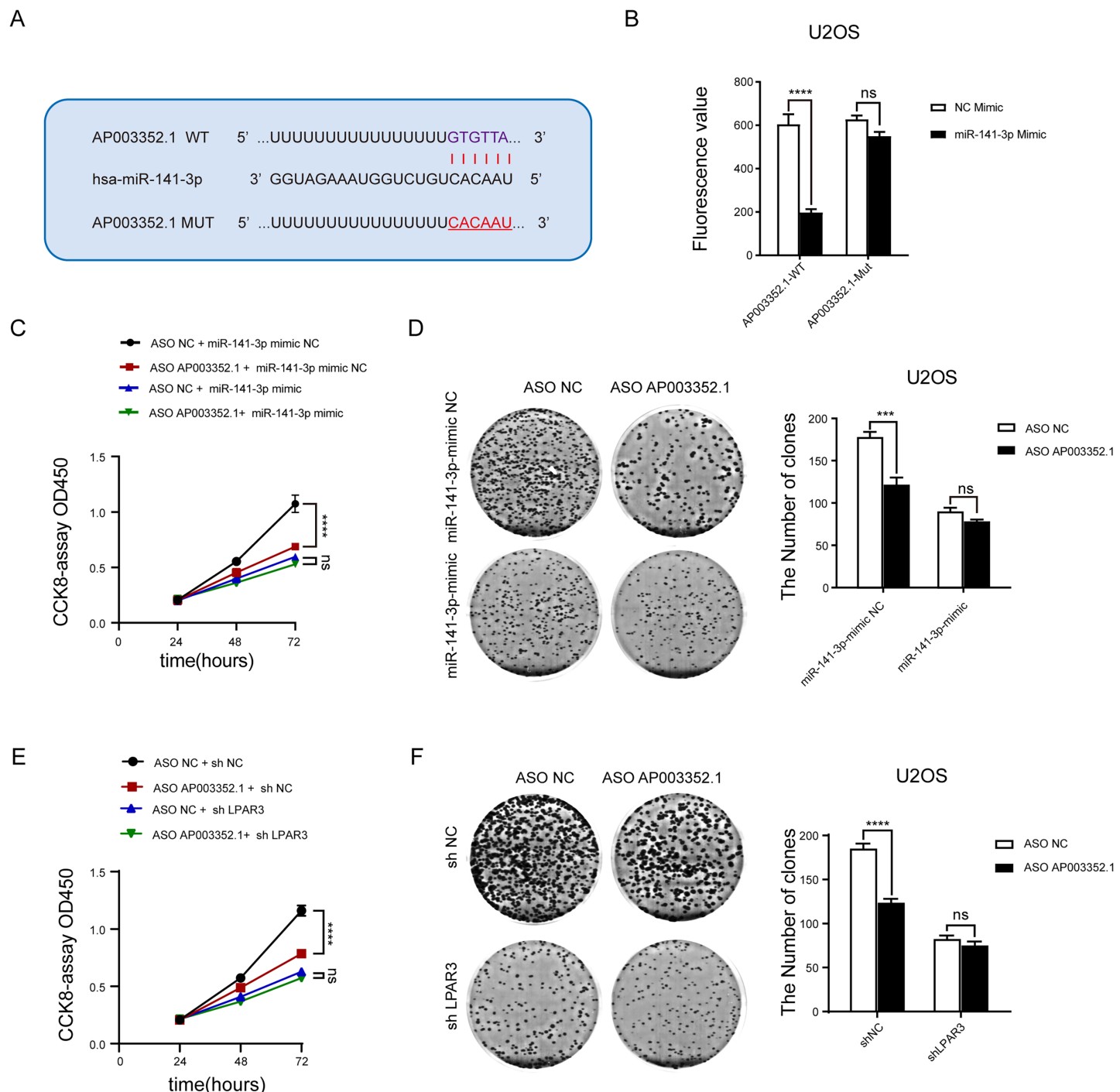

**Figure 7 As the ceRNA of miR-141-3p, AP003352.1 regulates LPAR3 to affect the malignant progression of osteosarcoma.** Predicted binding site between miR-141-3p and AP003352.1 (A). Luciferase assays were performed to test the effect of miR-141-3p on wild-type or mutant AP003352.1 after treating with miR-141-3p mimic for 48 h (B). The CCK8 results of knockdown AP003352.1 in U2OS cells treated with NC mimic or miR-141-3p mimic for 48 h (C). The cloneformation results of knockdown AP003352.1 in U2OS cells treated with NC mimic or miR-141-3p mimic for 48 h (D). The CCK8 results of knockdown AP003352.1 in shNC or shLPAR3 U2OS cells (E). The cloneformation assay of knockdown AP003352.1 for 48 h in shNC or shLPAR3 U2OS cells (F). Values were expressed as the means ± SD from three experiments, and the asterisk indicates the statistical significance compared to the controls (***$p < 0.001$, ****$p < 0.0001$).

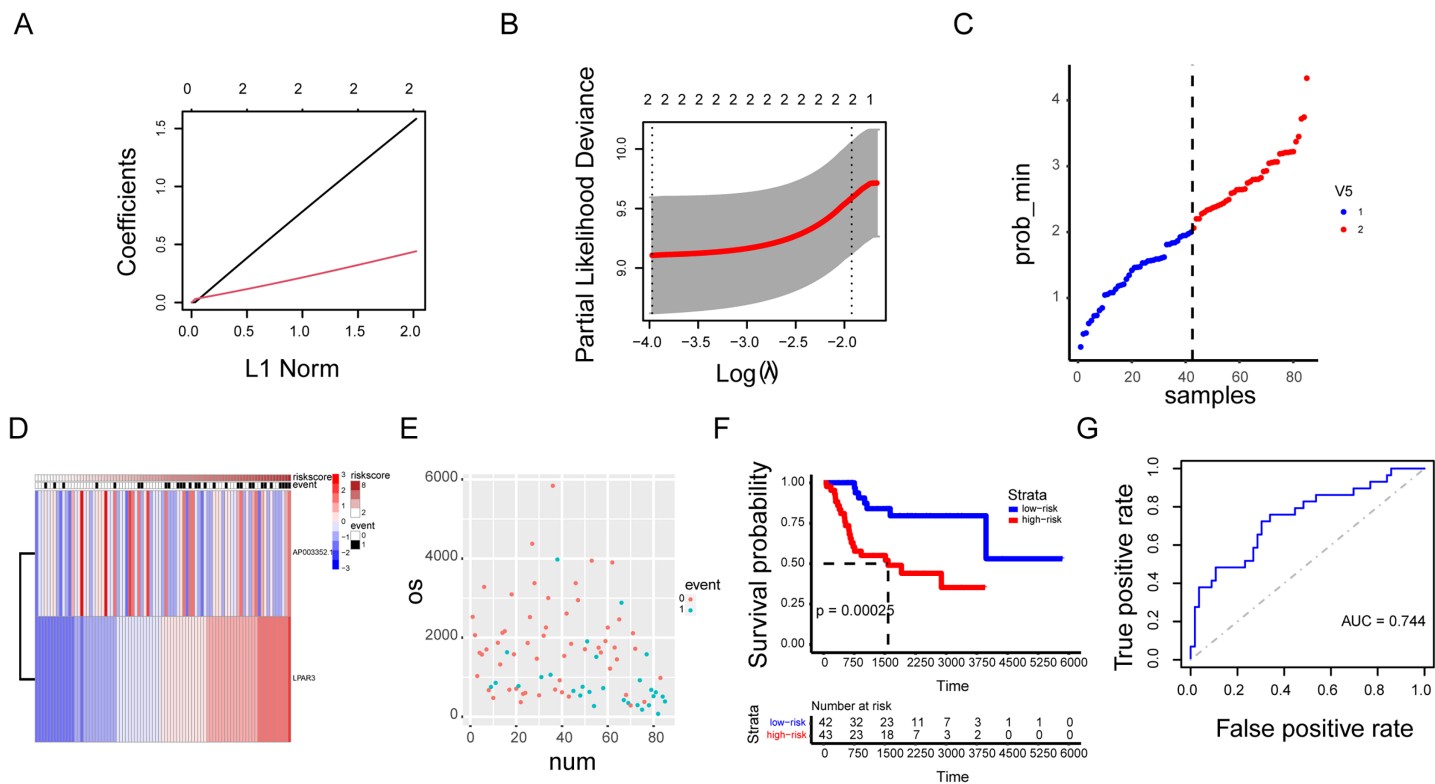

**Figure 8 The AP003352.1/miR-141-3p/LPAR3 axis can be a better biomarker for predicting osteosarcoma prognosis.** LASSO regression coefficient profile of the intersection genes (A). LASSO deviance profile of the intersection genes (B). Distribution of risk score between low and high-risk groups in the training cohort (C). The heatmap based on the risk score in the training cohort (D). Survival status plot of the training cohort (E). Survival curves for the two groups in the training cohort (F). ROC curves based on the risk score in the training cohort (G).

(Figs. 7E and 7F). All above results showed that AP003352.1 regulated LPAR3 and affected the malignant progression of osteosarcoma, similar to the ceRNA of miR-141-3p.

## The AP003352.1/miR-141-3p/LPAR3 axis can be a better biomarker for predicting osteosarcoma prognosis

Our previous research results indicated that the three genes were correlated with the malignant progression of osteosarcoma (Figs. S6A and S6B). However, the malignant progression of osteosarcoma was co-regulated by a multi gene network. Therefore, there were some limitations in the diagnosis of osteosarcoma by using a single gene as the biomarker. A prognosis model based on AP003352.1/LPAR3 expression was established through LASSO regression analysis (Figs. 8A–8D, hazard system = LPAR3 × 0.3996556 + AP003352.1 × 1.4405687). In addition, higher risk coefficients were associated with poorer survival, and the AUC values in both the training set and validation set were significantly higher than those in the single gene model (Figs. 8E–8G and Figs. S6C–S6F). Thus, we constructed a novel prognosis model by AP003352.1, miR-141-3p and LPAR3 genes in osteosarcoma.

## DISCUSSION

Osteosarcoma is a tumor that is highly malignant and has a poor prognosis. There is currently no effective treatment for osteosarcoma. Therefore, new diagnosis and treatment methods are urgently needed for patients with osteosarcoma. Meanwhile, LPAR3 is a crucial factor in osteosarcoma. With the development of bioinformatics, an increasing number of molecular markers can be used to predict the prognosis of tumors. However, complex gene networks are still needed to regulate the occurrence and development of tumors. Biomarkers are of great value, but there is no consensus on the basic definitions and concepts involved clinical practice and research. In addition, there are limitations in understanding the complexity of biomarkers in chronic diseases (*Califf, 2018*). Therefore, we need to conduct an in-depth exploration of the regulatory network of LPAR3 in osteosarcoma and establish a polygene diagnostic model. Non-coding RNAs, such as lncRNA and microRNA, play a significant role in osteosarcoma. LPAR3 is also closely related to non-coding RNA. Previous studies have shown that the interaction between LPAR3 and drugs could be predicted by constructing the lncRNA-miRNA-mRNA system grid, which play an essential role in the development of ischemic stroke (*Sun et al., 2021*). In addition, in prostate cancer, microRNAs can predict miRNAs that regulate hub genes. Through the analysis of an mRNA data set, it was found that LPAR3 was involved in the proliferation of prostate cancer cells. The miRNA of LPAR3 could be used as a marker for the detection of high-level prostate cancer, which was linked to the low survival rate and treatment resistance of patients with prostate cancer (*Foj & Filella, 2019*). However, the ceRNA regulatory network of LPAR3 has not been explored in relation to osteosarcoma. Our study indicated that AP003352.1 regulated the expression of LPAR3 in osteosarcoma, similar to the ceRNA of miR-141-3p. The above research established a co-regulatory network of LPAR3 and non-coding RNA in osteosarcoma, which provided a theoretical basis for targeted LPAR3 diagnosis and treatment. Our research has greatly enriched the regulatory network of ceRNA in osteosarcoma.

Lysophosphatidic acid receptor 3 (LPAR3) is a type of heterotrimeric G protein-linked receptors for lysophosphatidic acid (*Ishii et al., 2004*). It was reported that LPAR3 could be coupled with G protein to regulate a variety of signal pathways, such as the PI3K/Akt pathway and Ras pathway, which affected cell proliferation, cell apoptosis, and other functions (*Chun & Rosen, 2006*). It was reported that LPAR3 is a biological key point in the pathogenesis of tumor cells and participate in the regulation of tumorigenesis (*Hwang, Kim & Lee, 2022*). The expression form and level of LPAR3 were different in various malignant tumors, but LPAR3 often played a positive regulatory role in tumorigenesis. In addition, LPAR3 also participated in the regulation of osteosarcoma and had a key function in regulating the proliferation of osteosarcoma cells. LPAR3 could induce the proliferation and affect the viability of osteosarcoma cells. Moreover, LPAR3 could also promote the progression of thyroid cancer through the ceRNA axis, which promoted cell proliferation and inhibited cell apoptosis (*Xia & Jie, 2020*). However, the ceRNA network of LPAR3 has not been reported in osteosarcoma. All the above studies have established a

co-regulatory network of LPAR3 and non-coding RNA in osteosarcoma, which proposed a theoretical basis for the diagnostic and treatment approaches targeting LPAR3.

With the developing bioinformatics, a vast range of molecular markers can be used to predict the prognosis of tumors. However, the occurrence and development of tumors are regulated by complex gene networks. Monogenic biomarkers still have some limitations in disease diagnosis. There are many factors affecting monogenic biomarkers, such as tissue specificity, stability *in vitro*, and sensitivity. Therefore, we need to further explore the regulatory network of LPAR3 in osteosarcoma and establish a multi-gene diagnosis model. Although there are many methods of establishing polygenic prognosis models, the accuracy and sensitivity of polygenic prognosis models are not satisfactory; thus, they cannot be applied in clinical practice. The emergence of LASSO artificial intelligence has resolved this problem and greatly increased the accuracy of disease prognosis models. LASSO has the characteristic of punishing the absolute value of the regression coefficient, which can reduce the variability and increase the accuracy of the model. In addition, LASSO can allow abundant variables in the model and automatically delete unnecessary variables when some coefficient reaches 0 (*Mceligot et al., 2020*). Our study predicted and verified the feasibility of using the AP003352.1/miR-141-3p/LPAR3 axis as a molecular marker through the LASSO model. The accuracy and sensitivity of a polygene prognostic model composed of three genes are significantly higher than those of a single gene prediction model composed of three genes.

As a molecular sponge of miR-141-3p, AP003352.1 competitively regulated the expression of an important oncogene LPAR3 in osteosarcoma. In the meantime, the AC124798.1/miR-141-3p axis may regulate the malignant progression of osteosarcoma through LPAR3. More importantly, the AP003352.1/miR-141-3p/LPAR3 axis can be used as a molecular marker to evaluate the prognosis of osteosarcoma.

### Funding

This research was funded by the Liaoning Provincial Natural Science Foundation of China, grant number 2019-ZD-1010 and 2023-MS-343, and the Dalian Science and Technology Innovation Foundation, Grant Number 2022JJ13SN088 and 2020JJ27SN087. The funders had no role in study design, data collection and analysis, decision to publish, or preparation of the manuscript.

### Grant Disclosures

The following grant information was disclosed by the authors:
Liaoning Provincial Natural Science Foundation of China: 2019-ZD-1010 and 2023-MS-343.
Dalian Science and Technology Innovation Foundation: 2022JJ13SN088 and 2020JJ27SN087.

## Competing Interests

The authors declare that they have no competing interests.

## Author Contributions

- Hongde Yu conceived and designed the experiments, analyzed the data, prepared figures and/or tables, authored or reviewed drafts of the article, and approved the final draft.
- Bolun Zhang conceived and designed the experiments, analyzed the data, prepared figures and/or tables, and approved the final draft.
- Lin Qi conceived and designed the experiments, analyzed the data, prepared figures and/or tables, and approved the final draft.
- Jian Han performed the experiments, analyzed the data, authored or reviewed drafts of the article, and approved the final draft.
- Mingyang Guan performed the experiments, authored or reviewed drafts of the article, and approved the final draft.
- Jiaze Li performed the experiments, prepared figures and/or tables, and approved the final draft.
- Qingtao Meng conceived and designed the experiments, authored or reviewed drafts of the article, and approved the final draft.

## Data Availability

The raw data is available in the Supplemental Files.

## Supplemental Information

Supplemental information for this article can be found online at http://dx.doi.org/10.7717/peerj.15937#supplemental-information.

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
