# Peer review of "AP003352.1/miR-141-3p axis enhances the proliferation of osteosarcoma by LPAR3"

_PeerJ, doi:10.7717/peerj.15937_

## Round 0.1 · original submission · Major Revisions

Please address the reviewer comments and resubmit at the earliest

Reviewer 1 ·

Basic reporting

Ambiguous description of several critical experiments.

For example,
what is 'survival analysis' (line 224)?

What is ASO knockdown (line 172)?

I'm assuming a lot of R, Python packages have been used for data analysis, however no reference of the open-sourced tools are included.

Experimental design

Lack of detailed description for CCK8 and clone formation assay, which are critical experiments to demonstrate how target genes may impact cell proliferation.

To support hypothesis, more than one cell line should be tested to avoid artifact.

Validity of the findings

Findings from the article do support that AP003352.1/miR-141-3p can have impact on proliferation of osteosarcoma by LPAR3. However, the correlation is relatively weak, thus need more evidence to verify.

Reviewer 2 ·

Basic reporting

- Additional language edits are recommended to enhance the clarity and readability of the scientific discussions.
- Line 41-42: “MicroRNAs are noncoding RNAs that have been greatly conserved in evolution, consisting of approximately 22 amino acids.” – MicroRNAs do not contain amino acids. Do the authors mean nucleotides?
- The introduction could be improved by including more insights in place of general statements, like lines 45-47: “Abnormal expression of microRNAs is closely related not only to the pathogenesis and tumorigenesis, but also to the pathogenesis of many diseases”
- The results were overall presented with insufficient descriptions/details in the main text. For example, in Figure 1A-C, the only description was line 306 “To explore the internal mechanism, we performed differential expression analysis and univariate Cox analysis on mRNAs, lncRNAs, and miRNAs of osteosarcoma samples and precancerous samples, respectively (Figure 1A-C and Supplementary Figure 1).” – this section could be improved by describing the key findings from the figures instead of just presenting what analysis was done.
- Supplemental figures were also not described in detail. Key findings are not presented or integrated adequately in the main text. It is suggested to check the proper integration of the supplemental figures into the main text.
- Supplemental Figure S1 seems to be missing labels (mRNAs, lncRNAs, and miRNAs).
- Text in Figure 2B-C should be made bigger.
- For Figure 3A. it could be beneficial to highlight the axis described in Figure 3B. There are many colors that are quite similar to one another, and it was difficult to keep track of them - it would help to point them out better.
- Figure 4B labels were not legible. It would help to make the text bigger.
- For all Figures reporting a statistical test, please indicate the statistical tests used in the Figure caption. i.e., Figure 4C (Pearson’s?), Figure 4D (t-test?), etc., and indicate the significance level annotations in the Figure caption – define the respective significance level of “*”, “**”, “***” etc.
- Please provide information on the source/vendor of the ASOs. Only information on AP003352.1 was provided.
- Figure 4C: most of the p-values do not suggest significance. It may be beneficial to provide a bit more discussion on these results.
- It is suggested to quantify the western blot results in Figure 4E, as the difference between them is not very clear, especially considering there are also some fluctuations in the loading control.
- Was any particular normalization for CCK8 assays done? The cell viability was all the same at 24h – was this a raw measurement, or if the quantification were in some way normalized to the 24h time point?
- For Figures 5-7, please indicate the number of independent experiments or biological replicates performed in the Figure caption.
- For all bioinformatics analyses conducted in this study, it would be beneficial to provide additional details to ensure the reproducibility of the results. Specifically, please include the version numbers of the software packages used, any new codes developed to generate the figures, and the databases accessed and used in the study. This allows for better transparency and enhances the reproducibility of the work.

Experimental design

- It might help with reproducibility to add the experimental methods for lentiviral transduction for the generation of the stable shRNA cell line.
- There does not seem to be data presented to show the validation of each ASO knockdown in Figure 4D-E. All knockdowns need to validate for knockdown efficiency. I suggest using qRT-PCR.
- Please indicate the treatment time in the Figure caption for each of the conditions in Figure 5C, Figure 6-7.
- It is recommended to validate the effects of each of the ASO AP008853.1, miR-131-3p inhibitor, and shLPAR3 for the same duration (time to endpoint) as the experiments reported in Figure 5 to confirm the transfection efficiency and that the phenotypes correspond to the described treatments.
- Line 173-174: “The inhibitors, mimics, and ASOs of miRNA were transfected into U2OS cells using Lipofectamine 3000”. It appears that this is a transient transfection. In that case, would the effects of the transfection last for the entire duration of the experiment? Particularly the colony formation experiment, which is up to 10 days (according to line 143). Validation experiments would be beneficial – see comment above. Cell lines with stable expressions are generally more appropriate for experiments concerning extended treatments.
- For knockdowns, it is generally recommended to have two target sequences. This will greatly enhance the reproducibility of the results.

Validity of the findings

- Line 211-212: “The C5-GO enrichment analysis results demonstrated that these genes were mainly involved in the proliferation function of osteosarcoma” – proliferation was only one of the pathways in the GO term analysis. It is neither the most significant nor the most significant count or ratio. The claim that most of the genes are involved in proliferation is not fully justified.
- Figure 3B presents one axis that may be possible. However, from Figure 3A, there is an alternative axis. Please provide more justification for the selection of the particular axis. Also, it is recommended not to make significant interpretations yet on the implications of the findings, as these findings will still need to be validated.
- “Among mRNAs, LPAR3 is important in the regulation of osteosarcoma.” – it would be beneficial if the authors could present more detailed evidence or cite relevant literature that supports the significant role of LPAR3 in osteosarcoma.
- Line 226-227: “The results of differential expression analysis showed that six types of lncRNAs were significantly overexpressed in osteosarcoma (Figure 4A)” – It appears that AC124798.1 is not overexpressed, according to Figure 2A. Also, please report any statistical tests performed to justify the significance.
- The data presented in Figure 4D-E may require additional validation, as there weren’t sufficient data presented to show that the knockdowns were successful. See the Experimental Design section.
- Line 269-270 “The results showed that there was significant gene enrichment in the proliferation pathway (Supplementary Figure 5)”. However, note that the proliferation-related pathways highlighted, as shown in Figure S5, are related to smooth muscular cell, immune cell, and endothelial cell proliferation and may not have direct relevance to osteosarcoma, especially in the context of this study. The biological significance of these pathways in osteosarcoma should be clarified.
- A 10-day colony formation assays appear to be performed with transient transfection. Without validation that the effects of the transfection can persist for up to 10 days, the results may not be sufficiently robust. See the Experimental Design section.
- Line 287-288: “Thus, the prognosis model constructed by AP003352.1, miR-141-3p, and LPAR3 genes can more accurately diagnose osteosarcoma” – consider rephrasing this statement, as the argument made in this section only surrounds that these genes are potentially molecular predictors, suggesting a possible prognosis model – and not directly related to "accuracy" in diagnosis.

Additional comments

Overall, this is a very interesting manuscript. However, there are some limitations to reporting, experimental design, and validity of findings that need to be addressed. Comments with suggestions are provided above. Please consider spending a bit more time describing and explaining the findings presented in the figures – as this would help with the clarity of the manuscript.

---

## Round 0.2 · Minor Revisions

Please revise at the earliest.

Reviewer 1 ·

Basic reporting

Basic reporting has been significantly improved.

Experimental design

Key findings have been better described. Additional experiments have been supplemented to the manuscript.

Validity of the findings

Validation of finding have been improved, now evidence better support the statements.

Reviewer 2 ·

Basic reporting

The manuscript has significantly improved following the first round of revisions.
Comments on basic reporting from the previous round of review were mostly addressed.
Below are some additional minor comments:
- Figure 7D’s quantification label appears to show “No. of migrated cells” for a colony formation assay – please check and revise as necessary.
- Figure 6A & 7F treatment time was not indicated in the Figure caption.

Experimental design

Comments on experimental design were adequately addressed.

Validity of the findings

Comments on the validity of findings were mostly addressed.
Below are some additional comments.
- Line 298-299: “The results showed that in the control group, AP003352.1 knockdown inhibited the proliferation of osteosarcoma cells U2OS, while this inhibition was not observed in the mimic group (Figure 7C, D).” – it appears that Figure 7C is a proliferation assay while 7D is a colony formation assay. It may not be appropriate to equate proliferation and colony formation. Please revise as necessary.
- The comment on the first round of review: “Line 211-212: “The C5-GO enrichment analysis results demonstrated that these genes were mainly involved in the proliferation function of osteosarcoma” – proliferation was only one of the pathways in the GO term analysis. It is neither the most significant nor the most significant count or ratio. The claim that most of the genes are involved in proliferation is not fully justified” was not adequately addressed. Please provide more support in the section on lines 229-233.
- In the rebuttal letter, the authors mentioned the following for a comment on supplemental Figure 5 that the pathways identified are not directly related to osteosarcoma: “The genes were related to proliferation pathways in other cells, which suggested that the genes were indirectly related to osteosarcoma proliferation and had certain significance”. The response is appreciated, but it is recommended to articulate this similarly in a revised manuscript (around lines 295-296) to avoid misinterpretations.

---

## Round 0.3 · accepted · Accept

The paper has been revised to satisfaction.